# Epidemiology of Uveitis from a Tertiary Referral Hospital in Bulgaria over a 13-Year Period

**DOI:** 10.3390/diagnostics15070828

**Published:** 2025-03-25

**Authors:** Vesela Todorova Mitkova-Hristova, Marin Anguelov Atanassov, Yordanka Mincheva Basheva-Kraeva, Velichka Zaharieva Popova, Krasimir Iliev Kraev, Steffanie Hristova Hristova

**Affiliations:** 1Department of Ophthalmology, Faculty of Medicine, Medical University of Plovdiv, Clinic of Ophthalmology, University General Hospital “St. George”, 4001 Plovdiv, Bulgaria; 2Department of Propedeutic of Internal Diseases, Faculty of Medicine, Medical University of Plovdiv, Clinic of Rheumatology, University General Hospital “Kaspela”, 4001 Plovdiv, Bulgaria; 3Department of Propedeutic of Internal Diseases, Faculty of Medicine, Medical University of Plovdiv, Clinic of Rheumatology, University General Hospital “St. George”, 4001 Plovdiv, Bulgaria; 4Faculty of Medicine, Medical University of Plovdiv, 4002 Plovdiv, Bulgaria

**Keywords:** uveitis, epidemiology, HLA B27-associated uveitis, systemic association

## Abstract

**Objectives:** The aim of this study was to establish the etiology of uveitis and to examine its relationship with anatomical localization, age, and gender. **Methods:** A prospective study on patients with uveitis was conducted over a 13-year period at the Department of Ophthalmology, University Hospital “St. George”, Plovdiv, Bulgaria. Each case was diagnosed based on a comprehensive eye examination, a review of the systems, and additional laboratory and specialized examination methods. Patients were categorized into four groups based on the location of inflammation: anterior uveitis, intermediate uveitis, posterior uveitis, and panuveitis. **Results:** A total of 606 patients with uveitis were included in the study. The mean age of the study group was 46.5 ± 18.6 years. There was no statistically significant difference in gender distribution (*p* = 0.329). Anterior uveitis was the most dominant anatomical localization (*p* < 0.001). Cases with clarified etiology were significantly prevalent (*p* < 0.001). The most frequently identified etiology was HLA B27-associated uveitis (32.5%), followed by viral uveitis (16.8%). A significant correlation between etiology and anatomical localization was found (*p* < 0.001). The highest proportion (93%) of cases with clarified etiology was associated with posterior uveitis, while the lowest (39.7%) was linked to intermediate uveitis. **Conclusions:** We found that anterior uveitis was the most common anatomical localization, followed by intermediate uveitis. The disease is rare in childhood, while in elderly patients, there is an increase in idiopathic and viral uveitis cases. Our results provide valuable information about the most common etiologies of uveitis among the Bulgarian population.

## 1. Introduction

Uveitis is a global issue, with an incidence ranging from 17 to 52 cases per 100,000 people annually [1,2,3]. It can occur in any age group but primarily affects working-age adults (20–60 years) [3]. Inflammatory diseases of the uvea often recur, sometimes involving both eyes, and can lead to permanent vision impairment and reduced quality of life. Uveitis accounts for 10% of irreversible blindness in Europe and the USA and 25% worldwide [4], with its incidence having tripled in the past decade. As a result, the condition carries significant socio-economic implications [5] and early detection and timely treatment are therefore crucial.

Diagnosing uveitis correctly is always a challenge for ophthalmologists. The clinical patterns vary in different populations depending on ethnic, environmental, and socio-economic factors. Modern and continuously evolving diagnostic tools play a key role in the improved etiological clarification of uveitis and the identification of more specific diagnoses [4]. The clinical presentation of uveitis has also undergone evolutionary development due to emerging diseases and new surgical procedures. All of these facts necessitate the continuous update of epidemiological data.

The aim of this study was to determine the etiology of uveitis in patients who underwent treatment at the Department of Ophthalmology, University Hospital “St. George”, Plovdiv, Bulgaria, over a 13-year period, as well as to explore its association with anatomical localization, age, and gender.

## 2. Patients and Methods

A prospective observational study was conducted on patients with uveitis who were treated at University Hospital “St. George” in Plovdiv, Bulgaria, over a 13-year period from January 2011 to December 2023. The research protocol was reviewed and approved by the scientific ethics committee at Medical University—Plovdiv, Bulgaria (approval code No. 9, approval date: 13 May 2010). All procedures were carried out in accordance with the World Medical Association Declaration of Helsinki (1964) and its 2000 revision (Edinburgh). All participants provided written informed consent for the use of their data in scientific publications.

### 2.1. Inclusion and Exclusion Criteria

The study included patients diagnosed with endogenous uveitis and evidence of active inflammation, as well as those with a “masking syndrome”, initially identified as intraocular inflammatory processes. Demographic data (age at first symptoms, gender, and race), along with clinical data (anatomical localization, etiological diagnosis, laterality), were recorded. In cases of uveitis recurrence, the age and time from the first symptoms were considered the first episode, and these data were included in the study.

Patients were excluded from the study if they had: (a) inflammatory diseases affecting the adnexa and/or ocular coverings and contents in combination with the uveal tract, such as inflammatory diseases of the orbit, keratitis, scleritis; (b) a past inflammatory disease of the uveal tract with no current activity of uveitis at the time of the study; (c) secondary uveitis observed following trauma, surgical intervention (glaucoma surgery, cataract extraction and/or vitrectomy), or intravitreal medication administration.

### 2.2. Variables of Research Interest

#### 2.2.1. Etiology of Uveitis

The etiology of uveitis was a primary variable of interest. All patients underwent a complete ophthalmological examination, which included assessment of visual acuity, biomicroscopy, applanation tonometry, and indirect ophthalmoscopy. Additionally, based on the physician’s judgment, additional ophthalmological tests were performed, such as optical coherence tomography, fluorescein angiography, computerized visual field testing, and B-scan ultrasonography. Each patient also underwent a minimal laboratory screening that included a complete blood count, differential blood count, erythrocyte sedimentation rate, creatinine, urea, uric acid, urine analysis, and serological testing for syphilis.

In cases suspicious of an underlying systemic disease, the uveitis specialist (V.M-H) performed additional laboratory, radiological, and other investigations beyond the protocol, depending on the anatomical localization and clinical findings. These patients were referred for consultation with respective specialists (infectiologist, rheumatologist, pulmonologist, neurologist, nephrologist, phthisiatrist, pediatrician), and appropriate additional tests were conducted to confirm or exclude the suspicion of extraocular systemic involvement.

Based on etiology, patients were categorized into seven groups. The first six groups included uveitis with clarified etiology, while the last group consisted of all idiopathic cases. We grouped together patients with HLA-B27-positive acute anterior uveitis without systemic diseases and those with uveitis accompanied by ankylosing spondylitis, reactive arthritis, psoriatic arthritis, chronic ulcerative colitis, and Crohn’s disease. For the sake of brevity, we will hereafter refer to them as HLA-B27-associated uveitis/HLA B27+.

The second group comprised all cases with viral etiology, including herpes simplex (HSV), herpes zoster (HZV), cytomegalovirus retinitis (CMV), acute retinal necrosis (ARN), Posner–Schlossman syndrome, and Fuchs uveitis syndrome. As testing intraocular samples for viral pathogens was not possible, the diagnosis relied on typical clinical findings, systemic serological tests, and response to antiviral therapy. Due to the inability to test for viral pathogens from intraocular samples, the diagnosis was made based on typical clinical findings, systemic serological tests, and the response to antiviral therapy. Due to a sufficient number of observations, patients with juvenile idiopathic arthritis (JIA), rheumatoid arthritis (RA), and toxoplasmic retinochoroiditis (TRC) were categorized into a separate group, while other less frequently observed etiological diagnoses (with a frequency of one to ten cases) were included in the “rare diagnoses” group. This categorization was made to facilitate statistical analysis and was aligned with the frequency of etiological causes in our sample.

The diagnostic criteria for the most representative types of uveitis in our study are briefly described below. Acute anterior uveitis was considered to have a herpetic infection when it was unilateral, recurrent, associated with sectoral iris paralysis or atrophy, and/or a history of herpetic keratitis or herpes zoster ophthalmicus observed in the affected eye, accompanied by elevated intraocular pressure. HLA-B27-associated anterior uveitis was diagnosed based on acute, non-granulomatous, recurrent, alternating anterior uveitis, which tested positive for HLA-B27 antigen carriage. All cases of HLA-B27-positive uveitis were further evaluated by a rheumatologist to confirm the disease according to current rheumatological criteria. The diagnosis of Crohn’s disease and ulcerative colitis was confirmed by a gastroenterologist.

Patients with uveitis caused by JIA were usually diagnosed with arthritis before uveitis. However, in rare cases, uveitis was the first manifestation of JIA. For these patients, specific serological tests and consultations with a pediatric rheumatologist were conducted. For patients with RA and uveitis, the necessary laboratory screenings and consultations with a rheumatologist were also performed. Toxoplasmic retinochoroiditis was diagnosed through ophthalmoscopic examination, characterized by typical clinical findings: an atrophic scar with hyperpigmented edges, alongside fresh yellow-cream lesions indicative of active retinitis, and inflammatory exudation within the vitreous body.

In all cases of TRC, the diagnosis was confirmed by positive IgG titers for Toxoplasma. For patients suspected of tuberculosis-associated uveitis, chest radiography or computed tomography, QuantiFERON-TB, or T-SPOT.TB tests and consultations with a phthisiatrist were performed. Due to the high cost of gamma-interferon tests in our country, a two-step diagnostic approach was used for patients who were unable to afford them. The first step included a Mantoux tuberculin skin test, chest radiography or computed tomography, and consultation with a phthisiatrist. If the skin test was positive and the phthisiatrist determined it necessary, the patients would then undergo a gamma-interferon test.

The diagnosis of uveitis associated with Behçet’s disease followed the International Criteria for Behçet’s Disease [6] and the SUN Working Group Classification Criteria for Uveitis in Behçet’s Disease [7]. For cases of uveitis associated with sarcoidosis, histological confirmation was mandatory. However, two out of five patients did not undergo biopsies due to their personal refusal of the procedure. These cases were classified as presumptive and probable ocular sarcoidosis based on the revised diagnostic criteria adopted at the International Workshop on Ocular Sarcoidosis (IWOS) [8].

In cases of suspected multiple sclerosis (MS), the revised McDonald criteria were applied in collaboration with a neurology expert. Patients with HIV infection and syphilis were categorized together. For patients with a positive nonspecific or classical syphilis reaction, the diagnosis was confirmed using an enzyme-linked immunosorbent assay (ELISA) and consultation with a dermatologist. In cases of uveitis caused by Lyme disease, Boutonneuse fever, Cat scratch disease, or HIV, the etiology was confirmed through serological testing by an infectious disease specialist.

Toxocariasis in the patient with posterior uveitis was diagnosed based on a positive specific serological test performed by a parasitologist. Patients with uveitis and systemic lupus had a previously confirmed systemic disease, while those with tubulointerstitial nephritis-uveitis syndrome were diagnosed following a renal biopsy. White dot syndrome in four participants was identified based on characteristic ophthalmoscopic findings and confirmed by specific imaging results.

In our study, we diagnosed Vogt–Koyanagi–Harada (VKH) syndrome in four patients, all of whom were HLA-DR4-positive carriers. Three had vitiligo, while one presented with headache and meningeal irritation. The patient with sympathetic ophthalmia had a history of perforating trauma to the other eye 15 years earlier, exhibited typical clinical findings, and responded well to cytostatic therapy. The only patient with intraocular lymphoma was diagnosed via vitreous biopsy.

Among patients with masquerade syndrome, the distribution was as follows: intraocular tumors were identified in two cases of anterior uveitis, non-Hodgkin’s lymphoma was diagnosed in one case of intermediate uveitis, and in the remaining five cases of panuveitis, malignant diseases in other organs were confirmed with secondary choroidal infiltration. Uveitis was classified as idiopathic when it was not linked to any systemic disease and did not correspond to a specific ocular syndrome that would allow classification into a well-defined etiological group, even after a thorough etiological investigation.

#### 2.2.2. Laterality of Uveitis

According to the laterality of uveitis, the patients were divided into two groups: unilateral and bilateral. A distinction was made based on whether both eyes were simultaneously involved in the inflammatory process (number of eyes affected simultaneously) or whether the inflammatory process initially affected one eye and subsequently involved the other eye over time (number of affected eyes).

##### Anatomical Localization

Another variable of interest was the anatomical localization of the inflammatory process. According to the classification proposed by the International Uveitis Study Group (IUSG) [9], which is based on the primary anatomical localization of uveitis, the patients were divided into four groups: those with anterior uveitis, intermediate uveitis, posterior uveitis, and panuveitis.

##### Demographic Variables

The demographic variables of interest in this study were gender and age. To determine the prevalence of uveitis by age, the patients were grouped into three age categories: under (≤) 18 years, 19 to 65 years, and over 65 years. This classification was made to determine the frequency of pediatric uveitis and uveitis in patients over 65 years of age. The research excluded ethnicity from consideration since all participants were Caucasian.

### 2.3. Statistical Methods

The data were analyzed using the statistical software program IBM SPSS Statistics for Windows, Version 27.0 (Armonk, NY, USA: IBM Corp.). The statistical methods applied in data analysis were tailored to the type of variables and the specific research objectives. Non-parametric statistical methods were predominantly used as most variables were categorical, ordinal, or binary. These variables were presented as counts and percentages (%). To assess dependencies among them, Fisher’s exact test was employed for dichotomous variables, while the chi-square test and z-test with Bonferroni adjustments were used for pairwise comparisons in target variables with multiple categories, such as age groups, anatomical localization, and etiology. For variables where some cells had expected counts less than 5, the analysis utilized the Monte Carlo approach with the chi-square test at a 99% confidence interval. The age of the participants was normally distributed according to the Kolmogorov–Smirnov test and it was described by the mean and standard deviation. All statistical tests were conducted with a 5% acceptable Type I error rate, and statistical significance was considered when *p* < 0.05.

## 3. Results

### 3.1. Demographic and Clinical Data

Out of 934 patients with uveitis who were treated at the Department of Ophthalmology, University Hospital “St. George”, in Plovdiv, Bulgaria, over a 13-year period, 606 met the inclusion criteria and participated in the study. The patients with uveitis ranged in age from 3 to 87 years, with an average age of 46.5 ± 18.6 years. The proportion of patients in the second age group, between 19 and 65 years, was significantly higher compared to the other two age groups (*p* < 0.001). The gender distribution showed a similar proportion of men and women, with no statistically significant difference (*p* = 0.329). According to anatomical localization, anterior uveitis significantly dominated (*p* < 0.001). A significantly higher number of cases of unilateral uveitis were observed (*p* < 0.001), as well as cases with clarified etiology (*p* < 0.001) (Table 1). Table 1 presents the distribution of patients with uveitis based on the etiological diagnosis and the anatomical localization of the inflammatory process.

The distribution of uveitis according to the etiological diagnosis and anatomical localization is presented in Table 2.

### 3.2. Analysis of the Relationship Between Etiology and Anatomical Localization of Uveitis

A significant relationship was found between etiology and anatomical localization based on the chi-square test (*p* < 0.001). The pairwise comparisons of the anatomical localizations using the z-test revealed that posterior uveitis was associated with the highest proportion of cases with clarified etiology, whereas intermediate uveitis showed the lowest proportion of cases with clarified etiology, with a significant difference between the two localizations (*p* < 0.001). The difference between posterior uveitis and anterior uveitis was also significant (*p* = 0.03). Idiopathic etiology was significantly associated with intermediate uveitis (60.3%) compared to the other three localizations (*p* < 0.001 for all comparisons) (Table 3).

Figure 1 illustrates the proportions of the seven etiology groups. HLA-B27-associated uveitis had the highest proportion (32.5%, *n* = 197) among the clarified diagnoses. Viral uveitis came next, accounting for a proportion of 16.8% (*n* = 102). The group of rare diagnoses accounts for 9.1% (*n* = 55), followed by 6.1% TRC (*n* = 37), 4.5% RA (*n* = 27), and 3% JIA (*n* = 18). The remaining 28.1% (*n* = 170) consists of idiopathic uveitis.

### 3.3. Age Groups and Etiology

Age groups showed a significant association with the types of etiologies (*p* < 0.001) (Table 4). HLA-B27-associated uveitis was most common in patients aged 19 to 65 years (38.6%) and least common in those over 65 years old (9.7%). Viral uveitis was most frequently observed in patients over 65 years old (35.4%) and least often in those under 18 years old (5%). Cases of TRC showed similar proportions in the age group under 18 years (7.5%) and between 19 and 65 years (7.3%), with the lowest percentage associated with patients over 65 years old (0.9%). RA was observed in 6.2% of the patients over 65 years and in 4.40% in the age group between 19 and 65 years, whereas JIA was most commonly found in the youngest age group (35%). Rare diagnoses did not show a specific association with the patients’ age. Patients over 65 years old had the highest relative proportion of idiopathic uveitis (38.1%), compared to 26.7% in the age group between 19 and 65 years, and 15% in patients under 18 years old.

### 3.4. Gender and Etiology

A significant association was found between the gender of the patients and the types of etiologies (*p* < 0.001). Pairwise comparisons using the z-test showed the following significant differences: (1) *HLA-B27-associated uveitis* was significantly more frequent in men (45.5%) compared to 18.5% in women. (2) In contrast, women observed *TRC*, *RA*, and *JIA* more frequently, albeit in smaller overall proportions. (3) *Idiopathic uveitis* was also more characteristic in female patients (33.6%) compared to 22.9% in men. (4) *Viral* and *rare uveitis* showed no significant differences related to the patients’ gender (Table 5).

### 3.5. Anatomical Localizations and Etiology

A significant association was observed between the types of etiology and the anatomical localization (*p* < 0.001) (Table 6). *HLA-B27-associated uveitis* was most commonly observed in anterior uveitis, accounting for 41.3% of all etiologies in this localization, compared to 3.4% in intermediate uveitis and 0% in the other two localizations. *Viral uveitis* had a higher relative proportion in panuveitis (21.2%) and anterior uveitis (19.3%) compared to 7% in posterior uveitis and 1.7% in intermediate uveitis. Cases with *TRC* represented the highest relative proportion in the posterior uveitis localization (60.5%) and were completely absent in anterior uveitis. Patients with *RA* and *JIA* most commonly presented with anterior uveitis, with *JIA* reaching statistical significance. *Rare diagnoses* were observed in panuveitis (36.4%), followed by posterior uveitis (25.6%) and intermediate uveitis (22.4%), and were least common in anterior uveitis (4%). *Idiopathic* uveitis had the highest relative proportion in the intermediate localization (60.3%).

## 4. Discussion

In this study, the average age of the overall uveitis group at presentation was 46.5 ± 18.6 years. Our results are consistent with published European and global data on the epidemiology and etiology of uveitis, where the average age of the examined patients ranges from 40.4 to 47 years [10,11,12,13,14]. Other reports from Europe, the Middle East, Egypt, India, and South America suggest a younger demographic, ranging from 30 to 39.1 years [3,4,15,16,17,18,19,20,21]. The older age of our study group is attributed to the presence of intraocular lymphoma cases, as well as other causes of uveitis masquerade syndromes. Additionally, we defined the onset of uveitis as the patient’s first hospitalization. Some patients may have experienced mild episodes managed in outpatient care without providing medical documentation. In our country, a system for medical data interoperability is still under development.

With a few exceptions [22,23], most uveitis studies from developed countries show that uveitis affects both men and women equally [4,10,11,12,13,16,17,20] or with a slight female predominance due to hormonal or immune-mediated sensitivity [12,14,19,21,24,25,26,27,28,29,30,31,32,33,34,35,36]. Similar results were observed in our group, where the male-to-female ratio was almost equal, with a slight male predominance (1.1:1; *p* = 0.32). In countries classified as high-income economies—Germany, the UK, Japan, etc.—women predominantly suffer from uveitis [26,27,28], while reports from developing countries, classified with low average income, show men as twice as numerous as women [3,15,18,29]. The reason for the predominance of men in developing countries is multifaceted. On one hand, access to healthcare between the sexes is unequal; on the other hand, socio-economic conditions expose men to higher disease risks, especially of an infectious origin [30].

The anatomical localization of uveitis is an important characteristic because it predicts the final visual acuity and the development of ocular complications such as cataracts, glaucoma, macular edema, etc. Most studies report that posterior uveitis has the highest frequency [11,12,13,14,17,20,21,31,32,33]. Our data corroborate these findings. Anterior uveitis was observed in 77.9% of cases, and this anatomical subtype significantly dominated compared to the other localizations, with a relative proportion slightly higher than the reported figures for Europe and the Middle East [26,31,34,35,36]. Given that primary care settings treat many mild cases, it is likely that the percentage of anterior uveitis is even higher. Data from tertiary centers do not represent a true sample of the general population; nevertheless, these studies are of great importance for determining the local epidemiological profile of uveitis.

Although most European reports suggest posterior uveitis as the second most frequent anatomical subtype [14,31,34], our results showed intermediate uveitis (9.6%) as slightly predominating over posterior uveitis (7.9%) among Bulgarian patients. The literature points out that intermediate uveitis is the rarest anatomical subtype [10,37], with its frequency ranging from 1.4% to 22% [10,34,38]. In German and Austrian cohorts, higher frequencies of intermediate uveitis have also been reported, with values of 22.9% and 14.8%, respectively [26,39]. This may be influenced by the widespread prevalence of MS and Fuchs’ uveitis syndrome in these regions. Similarly, studies by Borde et al. and Patil et al. reported intermediate uveitis as the second most common anatomical subtype, with rates of 31.9% and 31.8%, respectively [13,20].

We attribute these differences to the referral of complex cases to university clinics, which are third-level competence centers like ours, for in-depth diagnosis and treatment. In our sample, intermediate uveitis had the highest relative proportion of etiologically unclear cases (60.3%). Very similar data have been reported by Rajan et al. (64.7%) [11] and Brydak-Godowska et al. (66.6%) [40] for idiopathic intermediate uveitis.

Literature sources indicate that the frequency of posterior uveitis varies widely. In Brazil and India, posterior uveitis is the leading anatomical subtype [10,18]. This is attributed to the widespread presence of infectious agents such as toxoplasmosis, tuberculosis, CMV, and onchocerciasis in these regions. On the other hand, posterior uveitis is associated with more severe visual impairment. In our study, we diagnosed posterior uveitis in 93% of our patients, with toxoplasmosis being the leading cause. Other European authors, such as Grajewski et al. (Germany) and Barisani-Asenbauer et al. (Austria), have published very similar results, classifying posterior uveitis in 94% and 80.4% of their patients, with toxoplasmosis being the most frequent etiology in both studies [14,39].

Inflammatory diseases affecting the entire uvea were least common among our patients, accounting for 5.4% of all cases. Jakob et al. observed a similar rate (6.2%) of panuveitis in their cohort [26]. This anatomical subtype is more frequently observed in South America, Africa, Asia, and some Southern European countries. In these regions, panuveitis is the leading or second most common anatomical localization [12,15,16,41,42] due to the widespread presence of Behçet’s disease and Vogt–Koyanagi–Harada syndrome.

The progress in clinical research, the introduction of new and more advanced diagnostic tests, and the multidisciplinary approach in healthcare institutions explain the reduction in the number of idiopathic uveitis cases in recent decades. However, the proportion of undiagnosed cases remains significant in many countries, reaching 20–30% [15,17,19,31,40,43,44]. In our study, idiopathic uveitis accounted for 28.1% and was the second most frequent. Other recent publications have reported even higher percentages of idiopathic uveitis: Mercanti et al. (Italy) reported 44.4% [45], Jamilloux et al. (France)—49% [32], and Borde et al. (India)—48.1% [13]. The limited availability of specific diagnostic tools in some centers leads to a higher frequency of suspected idiopathic uveitis. According to the literature, more frequently unclassified anatomical forms are anterior uveitis and intermediate uveitis [13,31], and we also observed this association.

Although considerable efforts are made to determine the etiology of intermediate uveitis, these are often unsuccessful. In contrast, many cases of anterior uveitis have a mild course, without recurrences, and are not thoroughly investigated in practice [26].

In 71.9% of the patients with uveitis, we were able to establish the diagnosis, and we also found a significant association between etiology and anatomical localization. The most common etiology we observed was HLA-B27-associated uveitis (32.5%). Uveitis was characteristic of the age group 19–65 years in our cohort and was significantly more frequent in men. The prevalence of the HLA-B27 phenotype varies widely, from 15.9% in Norway to less than 1% in Japan [46]. It is believed that 30–80% of patients with seronegative spondyloarthropathies are HLA-B27 positive [26,47,48]. We found this association in 41.3% of the cases with anterior uveitis. Anterior uveitis is the first and most common extra-articular manifestation of the disease in 30–40% of cases, so in many instances, it precedes the rheumatological symptoms [48]. This requires genetic typing of patients with recurrent acute anterior uveitis.

Viral uveitis (16.8%) was the second most frequent etiology in our cohort. It is most common in the >65-year age group and least common in children (≤18 years). Risk factors for viral uveitis include advancing age, unilateral involvement, and the presence of keratouveitis [26]. Other authors also point to viral uveitis (HSV and VZV) as responsible for the largest number of cases of infectious uveitis [24,31].

Toxoplasmosis is a leading cause of posterior uveitis worldwide [4,21,26,40]. Cats are the definitive host of the parasite, while pigs and humans serve as intermediate hosts. Therefore, dietary and social factors significantly influence the geographic distribution of this infection. Toxoplasmosis is acquired through the consumption of food and/or water contaminated with oocysts or inadequately cooked infected meat. In Asian countries, the domestication of cats and the consumption of pork are not as widespread as in European countries, which accounts for the lower frequency of TRC in these regions [41]. In contrast, countries like Brazil, which report a high number of cases with infectious etiology, find toxoplasmosis to be a leading factor for uveitis. It is the third most common diagnosis (6.1%) and the most frequent cause of unilateral posterior uveitis (60.5%) in our study. All age groups are susceptible to toxoplasmosis, with a similar distribution in children and adults aged 19–65 years (7.5% and 7.3%, respectively), and a significant decrease in frequency after the age of 65 years (0.9%). In our study, women were more affected by the disease.

Other frequent causes of uveitis observed in our sample include patients with RA (4.5%) and JIA (3%). As expected, RA is more common in those over 65 years of age, while JIA is more prevalent in children. These diseases primarily affect the anterior segment of the eye and are more common in women, although with a slight difference from men. Other authors have also found higher frequencies for the female sex in these conditions [3].

Pediatric uveitis (≤18 years) accounted for 6.6% of our sample, aligning with previous studies that report frequencies ranging from 2.2% to 16% [10,11,18,49,50,51,52], with some authors even documenting higher rates [12,16]. In developed countries, the most commonly observed uveitis in children is associated with JIA, unlike traumatic uveitis and pars planitis, which are more common in developing nations [53]. In our study, we also found that the most frequent uveitis in the pediatric cohort was associated with JIA (35%), followed by juvenile HLA-B27-associated uveitis (27.5%), with inflammatory uveitis predominantly presenting as posterior uveitis (77.5% of the cases).

In our study, patients aged 65 years and older made up 18.6% of the total sample. Grajewski et al. (Germany) reported a similar proportion of 18% [14]. In a more recent report by Garcia-Aparicio et al. (Italy), the relative proportion for this age group reached 22.6% [12]. We observed that with increasing age, there was an increase in idiopathic (38.1%) and viral uveitis (35.4%), primarily due to herpes zoster (77.5% of all viral uveitis in the 65 and older age group). Many authors have found similar associations [14,17,26].

Table 7 summarizes the clinical features from related studies worldwide, which were referenced in our discussion of the results.

Determining the etiology of uveitis is essential for effective therapy and controlling the inflammatory process. In many cases, uveitis is associated with a systemic disease, necessitating interdisciplinary collaboration for accurate diagnosis and treatment. The diagnostic process begins with a detailed medical history, followed by a comprehensive eye examination and systemic evaluation to form a clinical conclusion and working diagnosis. Another critical aspect is considering the differential diagnosis and selecting the appropriate paraclinical, serological, and imaging tests to identify the cause. A non-selective approach to testing can be costly and ineffective, as many tests have low diagnostic value.

## 5. Limitations

One of the limitations is associated with the challenge of identifying the etiology of uveitis, which requires the use of appropriate diagnostic methods, resources, and equipment. However, in our country, certain procedures are not covered by mandatory health insurance, leading patients to cover the costs themselves. Additionally, laboratories sometimes lack the necessary resources and equipment for high-tech investigations, leading to a higher percentage of idiopathic uveitis cases. Moreover, as a tertiary center, our hospital primarily admits more severe cases, potentially resulting in an underrepresentation of milder forms of uveitis and an overrepresentation of rarer diagnoses. Another limitation arises from the fact that despite the study spanning a long period, it was conducted in a single hospital. Despite these limitations, this is the first contemporary, large-scale study of uveitis in our country.

## 6. Conclusions

In the present study, we found an equal gender distribution of uveitis, with anterior uveitis being the most common anatomical localization, followed by intermediate uveitis. Among the etiologically determined cases, HLA-B27-associated uveitis was the leading cause, followed by viral uveitis and TRC uveitis. The disease is rare in childhood, while in elderly patients, there is an increase in idiopathic and viral uveitis cases. Our results provide valuable information about the more common etiologies of uveitis among the Bulgarian population. The results have practical relevance for clinical practice in selecting suitable tests for determining the etiology of uveitis.

## Figures and Tables

**Figure 1 diagnostics-15-00828-f001:**
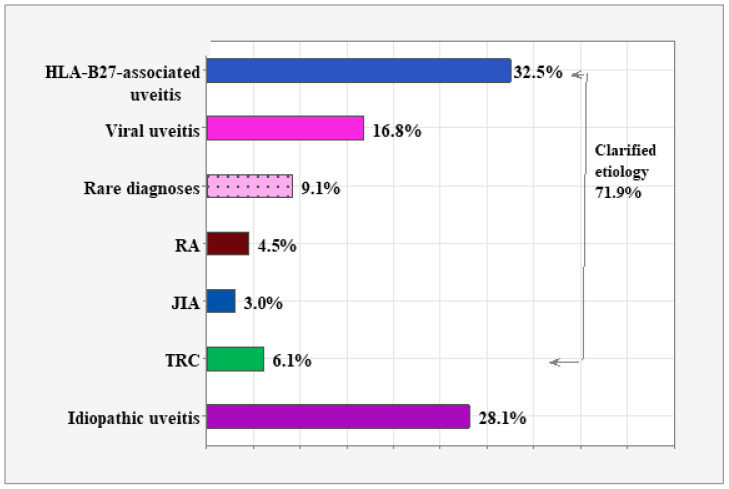
Distribution by etiological categories.

**Table 1 diagnostics-15-00828-t001:** Background information about the patients.

Variables	Statistics	*p*-Value
Age ○ mean (SD) ○ minimum–maximum	46.5 (18.6) 3–87	n. a.
Age groups ○ ≤18 years ○ 19 to 65 years ○ >65 years	40 (6.6%) ^a^ 453 (74.8%) ^b^ 113 (18.6%) ^a^	<0.001 ^z^
Gender *n* (%) ○ Men ○ Women	314 (51.8%) 292 (48.2%)	0.329 ^f^
Anatomic localization ○ Anterior uveitis ○ Intermediate uveitis ○ Posterior uveitis ○ Panuveitis	472 (77.9%) ^a^ 58 (9.6%) ^b^ 43 (7.1%) ^b^ 33 (5.4%) ^b^	<0.001 ^z^
Number of eyes affected at the same time ○ One eye affected ○ Both eyes affected	526 (86.8%) 80 (13.2%)	<0.001 ^f^
Number of consecutively affected eyes ○ One eye affected ○ Both eyes affected	429 (70.8%) 177 (29.2%)	<0.001 ^f^
Etiology ○ Clarified ○ Idiopathic	436 (71.9%) 170 (28.1%)	<0.001 ^f^

^f^ Fisher’s exact test; ^z^ z-test; Values with different letters in the same column indicate significant differences.

**Table 2 diagnostics-15-00828-t002:** Distribution of uveitis according to etiological diagnosis and anatomical localization of uveitis (*n* = 606).

Etiology	Anatomic Location				Total
	Anterior Uveitis	Intermediate Uveitis	Posterior Uveitis	Panuveitis	
Ankylosing spondylitis	93	1	0	0	94
Crohn’s disease	3	0	0	0	3
Intraocular lymphoma	0	1	0	0	1
Idiopathic uveitis	127	35	3	5	170
Lyme disease	0	2	0	0	2
Boutonneuse fever	0	0	1	0	1
Masquerade syndromes	2	1	0	5	8
MS	0	4	0	0	4
Sympathetic ophthalmia	0	0	0	1	1
Posner–Schlossman syndrome	13	0	0	0	13
Psoriatic arthritis	10	1	0	0	11
RA	24	2	0	1	27
Sarcoidosis	1	1	1	2	5
Behçet’s disease	3	0	1	3	7
Cat scratch disease	0	1	0	0	1
Reactive arthritis	36	0	0	0	36
Syphilis	1	0	0	0	1
Syphilis_HIV	1	1	0	0	2
SLE	0	2	0	0	2
Tuberculosis	7	0	1	0	8
TINU	3	0	0	0	3
Toxocariasis	0	0	1	0	1
Toxoplasmosis	0	4	26	7	37
Fuchs uveitis syndrome	6	0	0	0	6
HZV	41	0	0	0	41
HSV	31	1	3	1	36
UC	2	0	0	0	2
JIA	16	1	0	1	18
ARN	0	0	0	3	3
CMV	0	0	0	3	3
HLA B27+ nonsystemic	51	0	0	0	51
VKH	1	0	2	1	4
White dot Sy	0	0	4	0	4
**Total**	472	58	43	33	606

JIA—juvenile rheumatoid arthritis; RA—rheumatoid arthritis; HSV—herpes simplex virus uveitis; HZV—varicella zoster virus uveitis; SLE—systemic lupus erythematosus; CMV—cytomegalovirus uveitis; TB—tuberculosis; TINU—tubulointerstitial nephritis and uveitis; UC—ulcerative colitis; ARN—acute retinal necrosis; VKH—Vogt–Koyanagi–Harada syndrome; MS—multiple sclerosis.

**Table 3 diagnostics-15-00828-t003:** Distribution of the inflammatory process by etiology and anatomical localization.

Etiology	Anterior Uveitis	Intermediate Uveitis	Posterior Uveitis	Panuveitis	Chi-Square *p*-Value
Clarified	345 ^a^ (73.1%)	23 ^b^ (39.7%)	40 ^c^ (93%)	28 ^a,c^ (84.8%)	<0.001
Idiopatic	127 ^a^ (26.9%)	35 ^b^ (60.3%)	3 ^a^ (7%)	5 ^a^ (15.2%)	

The cells marked with the same letter are not significantly different from each other at the 0.05 level, based on z-test paired comparisons.

**Table 4 diagnostics-15-00828-t004:** Types of etiologies by age groups.

Etiology *n* (%)	≤18 Years	19–65 Years	>65 Years	Chi-Square Monte-Carlo *p*-Value
○ HLA B27+ ○ Viral ○ TRC ○ RA ○ JIA ○ Rare ○ Idiopathic	11 (27.5%) ^a^ 2 (5%) ^a^ 3 (7.5%) ^a^ 0 (0%) ^a^ 14 (35%) ^a^ 4 (10%) ^a^ 6 (15%) ^a^	175 (38.6%) ^a^ 60 (13.2%) ^a^ 33 (7.3%)^a^ 20 (4.4%) ^a^ 4 (0.9%%) ^b^ 40 (8.8%) ^a^ 121 (26.7%) ^a^	11 (9.7%) ^b^ 40 (35.4%) ^b^ 1 (0.9%) ^a^ 7 (6.2%) ^a^ 0 (0%) ^b^ 11 (9.7%) ^a^ 43 (38.1%) ^b^	<0.001

TRC—toxoplasmic retinochoroiditis; RA—rheumatoid arthritis; JIA—juvenile idiopathic arthritis; The cells marked with the same letter are not significantly different from each other at the 0.05 level, based on paired z-test comparisons.

**Table 5 diagnostics-15-00828-t005:** Types of etiologies by gender.

Etiology *n* (%)	Men	Women	Chi-Square *p*-Value
○ HLA B27+ ○ Viral ○ TRC ○ RA ○ JIA ○ Rare ○ Idiopatic	143 (45.5%) ^a^ 52 (16.6%) ^a^ 13 (4.1%) ^a^ 2 (0.6%) ^a^ 5 (1.6%) ^a^ 27 (8.6%) ^a^ 72 (22.9%) ^a^	54 (18.5%) ^b^ 50 (17.1%) ^a^ 24 (8.2%) ^b^ 25 (8.6%) ^b^ 13 (4.5%) ^b^ 28 (9.6%) ^a^ 98 (33.6%) ^b^	<0.001

TRC—toxoplasmic retinochoroiditis; RA—rheumatoid arthritis; JIA—juvenile idiopathic arthritis; The cells marked with the same letter are not significantly different from each other at the 0.05 level, based on paired z-test comparisons.

**Table 6 diagnostics-15-00828-t006:** Types of etiologies by anatomical localizations.

Variables *n* (%)	Anterior Uveitis	Intermediate Uveitis	Posterior Uveitis	Panuveitis	Chi-Square Monte-Carlo *p*-Value
**Etiology *n* (%)**					
○ HLA B27+ ○ Viral ○ TRC ○ RA ○ JIA ○ Rare ○ Idiopatic	195 (41.3%) ^a^ 91 (19.3%) ^a^ 0 (0%) ^a^ 24 (5.1%) ^a^ 16 (3.4%) ^a^ 19 (4%) ^a^ 127 (26.9%) ^a^	2 (3.4%) ^b^ 1 (1.7%) ^b^ 4 (6.9%) ^b^ 2 (3.4%) ^a^ 1 (1.7%) ^a^ 13 (22.4%) ^b^ 35 (60.3%) ^b^	0 (0%) ^b^ 3 (7%) ^b,c^ 26 (60.5%) ^c^ 0 (0%) ^a^ 0 (0%) ^a^ 11 (25.6%) ^b^ 3 (7%) ^c^	0 (0%) ^b^ 7 (21.2%) ^a,c^ 7 (21.2%) ^d^ 1 (3%) ^a^ 1 (3%) ^a^ 12 (36.4%) ^b^ 5 (15.2%) ^a,c^	<0.001

TRC—toxoplasmic retinochoroiditis; RA—rheumatoid arthritis; JIA—juvenile idiopathic arthritis; The cells marked with the same letter are not significantly different from each other at the 0.05 level, based on paired z-test comparisons.

**Table 7 diagnostics-15-00828-t007:** Summary of the clinical features from related studies worldwide.

Autor (Country) Time (Year of Publication)	Sample Size	Mean Age (Years)	Men to Women Ratio	AU (%)	IU (%)	PU (%)	Panuveitis (%)	Idiopathic Uveitis (%)	Etiologically Clarified Principal Causes (%)	Children Uveitis (%)	Elderly Uveitis (%)
Polania [4] (Columbia) 2013–2021 (2022)	489	38.7	1:1.7	45.8%	7.8%	16.3%	30.1%	16.6%	TRC (16%) HLA-B27+ (5.5%)	15.1% (<16 y)	38.5% (>50 y)
Rajan [11] (Malaysia) 2018 (2022)	1199	41.4	1.08:1	46.7%	11.3%	20.8%	21.2%	57.2%	TB (8.8%) Viral (8.9%) TRC (7.8%)	8.8% (≤20 y)	15.5% (>60 y)
Garcia-Aparicio [12] (Spain) 2016–2017 (2021)	389	47	1:1.3	81.5%	4.6%	3.9%	10%	48.4%	HLA-B27+ (13.4%) Viral (10.6%) JIA (2.8%)	8% (<18 y)	22.6% (≥65 y)
Borde [13] (India) 2016–2017 (2020)	210	46.6	1.03:1	47.1%	31.9%	12.9%	8.1%	48.1%	TB (11.9%) Viral (10%) HLA-B27+ (7.1%)	_	_
Grajewski [14] (Germany) 2012–2013 (2015)	474	45	1:1.2	53%	19%	21%	7%	41%	Viral (12.5%) Sarcoidosis (11.4%) HLA-B27+ (10.1%)	_	_
Abdelwareth [15] (Egypt) 2015–2017 (2021)	313	30	2.1:1	33.2%	0.6%	12.8%	53.4%	24.6%	Behҫet (29.1%) WKH (8.4%)	24.3% (<18 y)	2.9% (>60 y)
Amin [16] (Egypt) 2013–2016 (2015)	454	32	1.1:1	40.7%	7.3%	9%	43%	25.6%	Behҫet (28.6%) VKH (16%)	25.3% (≤16 y)	9.25% (≥60)
El Lativ [17] (Egipet) 2015–2020 (2019)	982	33.8	1.1:1	34.4%	20%	25.6%	20%	30.9%	TB (18.2%) Sarcoidosis (10.1%) Behҫet (8.2%)	33.8% (<18 y)	_
Pandurangan [18] (North India) 2019–2021 (2022)	102	39.1	1.6:1	23.4%	11.3%	46.8%	18.5%	56.7%	TB (18.5%) SO (7.1) HLA-B27+ (6.4%)	6.9% (<18 y)	9.8% (<60)
Hossseini [19] (Iran) 2013–2014 (2018)	235	35.8	1:1.5	37%	11.9%	4.3%	46.8%	28.5%	Behҫet (16.6%) VKH (10.6%)	_	_
Patil [20] (India) (2019–2020) (2023)	201	35.4	1.5:1	45.3%	31.8%	14.9%	8%	_	TB (8.5%)	_	_
De-la-Torre [21] (Columbia) 2010–2022 (2023)	3404	41.1	1:1.2	49.6%	5.2%	22.9%	22.3%	27.7%	TRC (25.3%) Viral (6.4%)	10.2% (≤19 y)	24.6% (≥60 y)
Kalogeropoulos [24] (Greece) 1991–2020 (2023)	6191	40.6	1:1.1	59.1%	5.9%	21.9%	13.1%	32.8%	Viral (34.1%) Sarcoidosis (9.8%) HLA-B27+ (8.2%)	3.9% (<18 y)	_
Jakob [26] (Germany) 2001–2006 (2008)	1916	35	1:1.3	45.4%	22.9%	13.5%	6.2%	35.3%	HLA-B27+ (14.2%) Viral (12.3%) TRC (4.1%)	_	_
De-la-Torre [29] (Columbia) 1996–2006 (2009)	693	31.7	1:1.1	28.9%	4.3%	35.9%	30.9%	39.8%	TRC (39.8%) Toxocariasis (6.3%)	19.6% (<16 y)	8.6% (>60 y)
Llorenç [31] (Spain) 2009–2012 (2015)	1022	45	1:1.2	52%	8%	23%	15%	26%	HLA-B27+ (24.7%) Viral (22.9%) TRC (7%)	7.24% (≤16 y)	_
Biziorek [34] (Poland) 1996–2000 (2001)	563	40.4	1:1.1	44.6%	7.3%	33.0%	15.1%	30%	_	_	_
Hamade [35] (Saudi Arabia) 1996–2007 (2009)	487	37	1.2: 1	60%	6%	11%	24%	10.5%	AAU (32%) Herpes virus (12%) TB (7%)	14.6% (≤18 y)	10.3% (>60 y)
Kazokoglu [36] (Turkey) 2004 (2008)	761	35.5	1.04:1	52.5%	6.7%	12.7%	28.1%	43.2%	Behҫet (32.1%) FUS (5.1%) TRC (4.7%)	6.3% (<16 y)	6.6% (>60 y)
Barisani-Asenbauer [39] (Turkey) 1995–2009 (2012)	2619	38.8	1:1.04	59.9%	14.8%	18.3%	7.0%	39.3%	HLA-B27+ (18.3%) Viral (8.6%)	11.7% (≤17 y)	13.5% (>61 y)
Brydak-Godowska [40] (Poland) 2005–2015 (2018)	279	38.3	1:1.6	26.5%	12.9%	48.4%	12.2%	23.7%	TRC (17.9%) FUS (12.2%)	_	_
de Moraes [44] (Brasil) 2002–2016 (2022)	408	42	1.1:1	39.5%	3.6%	52%	4.9%	22.5%	TRC (33.8%) JIA (6.1%)	8.6% (≤18 y)	7.6% (≥60 y)
Soheilian [53] (Iran) 1997–2000 (2004)	544	31.2	1:1.3	38.4%	17.6%	18.6%	25.4%	42.8%	TRC (10.1%) Behçet (8.6%)	14.9% (<16 y)	6.4% (>61 y)
Mitkova-Hristova [the present study] (Bulgaria) 2011–2023 (2025)	606	46.5	1.07:1	77.9%	9.6%	7.1%	5.4%	28.1%	HLA-B27+ (32.5%) Viral (16.8%) TRC (6.1%)	6.6% (≤18 y)	18.6% (>65 y)

AU—Anterior uveitis; IU—Intermediate uveitis; PU—Posterior uveitis; TB—tuberculosis, TRC—toxoplasmic retinochoroiditis, VKH—Vogt–Koyanagi–Harada, SO—sympathetic ophthalmia, AAU—acute anterior uveitis, FUS—Fuchs’ uveitis syndrome, JIA—Juvenile idiopathic arthritis.

## Data Availability

The data are available from the corresponding author upon reasonable request.

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
