# Peer review of "Epidemiology of Uveitis from a Tertiary Referral Hospital in Bulgaria over a 13-Year Period"

_diagnostics, 2025, doi:10.3390/diagnostics15070828_

Round 1
Reviewer 1 Report
Comments and Suggestions for Authors
Comment from a reviewer
The authors conducted a prospective study on 606 uveitis patients over a 13-year period, and the value of the study lies in the large sample size and comprehensive analysis. However, as noted below, there are several areas that require further clarification. Additionally, the manuscript would benefit from proofreading by a native English speaker.
L45 “Over the last few decades, there has been a significant shift in the ……..groups, from infectious to immune-mediated” The statement aligns with the findings of the cited paper from Columbia but does not reflect global or continental perspectives. The authors are encouraged to support their claim with a reference from a comprehensive literature review.
Table 1
- There is a typo in "19 to 65 years," along with other typographical errors. Please review and correct them.
- The column of “Statistics” should be revised, for examples, presenting mean + SD (range) for age
- I'm uncertain whether the use of the exclamation mark adheres to the author guidelines. Please review and ensure compliance.
L228 The current age grouping (19-65 years) is too broad for meaningful analysis. Could it be divided into smaller categories, such as 19-30, 31-45, and 46-65? This would provide more detailed insights, including the age group proportions for TRC and HLA-B27-associated uveitis.
Table 5 There is a typo in the title in column 1.
L275 “Other reports from Europe, …….ranging from 30 to 39.1 years” The reason for the observed difference should be clarified. Did the previous epidemiological studies include juvenile cases? Providing this information would help contextualize the findings.
L291 Repeated abbreviations such as PU, AU, IU, MS, etc., should be removed throughout the paper, along with other typographical errors. Please review and correct them.
Discussion To enhance this section, I recommend adding a newly created summary table for the literature review, as mentioned in the discussion paragraph. This table will help streamline the discussion and reduce the need for lengthy descriptions. Additionally, the manuscript currently lacks a section addressing the limitations of the study, which should be included.
Comments on the Quality of English LanguageThe English writing should be improved through proofreading by a native English speaker.
Author Response
March 3rd, 2025
Dear REVIEWER 1,
Thank you very much for the insightful comments and suggestions you have provided on our manuscript titled Epidemiology of Uveitis from a Tertiary Referral Hospital in Bulgaria over 13-Year Period. We have tried to implement all of them in the revision process. We have highlighted in yellow all sections, where additional information was provided or suggested revisions were done. All language edits, including grammar, word choice, rephrasing, and punctuation are given in red color. Here follows a point-by-point description of the revisions.
The authors conducted a prospective study on 606 uveitis patients over a 13-year period, and the value of the study lies in the large sample size and comprehensive analysis. However, as noted below, there are several areas that require further clarification. Additionally, the manuscript would benefit from proofreading by a native English speaker.
Response: We thank the reviewer for the time spent in reviewing our manuscript and the detailed critical feedback. We have tried to apply all suggested revisions in the process of improving our manuscript.
Comment 1: L45 “Over the last few decades, there has been a significant shift in the ……..groups, from infectious to immune-mediated” The statement aligns with the findings of the cited paper from Columbia but does not reflect global or continental perspectives. The authors are encouraged to support their claim with a reference from a comprehensive literature review.
Response: We agree with the reviewer that the statement does not reflect global or continental perspectives and we decided to remove it from the introduction on page 2.
Comment 2: Table 1
- There is a typo in "19 to 65 years," along with other typographical errors. Please review and correct them.
Response: We fixed this.
- The column of “Statistics” should be revised, for examples, presenting mean + SD (range) for age
Response: We added the minimum and maximums age in table 1.
- I'm uncertain whether the use of the exclamation mark adheres to the author guidelines. Please review and ensure compliance.
Response: We have changed this and marked the significant differences with superscript letters.
Comment 3: L228 The current age grouping (19-65 years) is too broad for meaningful analysis. Could it be divided into smaller categories, such as 19-30, 31-45, and 46-65? This would provide more detailed insights, including the age group proportions for TRC and HLA-B27-associated uveitis.
Response: We decided to divide the age groups in this way because we wanted to examine the frequency of pediatric and adult uveitis, similar to other studies. According to the Bulgarian healthcare system, patients are considered children until the age of 18, while the age of 65 and above are typically retired. The upper limit for the pediatric uveitis group varies between 16 and 20 years in different studies, while for adults, it is most commonly set at 60-65 years. This type of age grouping aligns with the practice in our country.
Comment 4: Table 5 There is a typo in the title in column 1.
Response: Thank you. This was fixed.
Comment 5: L275 “Other reports from Europe, …….ranging from 30 to 39.1 years” The reason for the observed difference should be clarified. Did the previous epidemiological studies include juvenile cases? Providing this information would help contextualize the findings.
Response: We elaborated on this issue in the discussion on page 10, Lines 305 – 310.
Comment 6: L291 Repeated abbreviations such as PU, AU, IU, PAN should be removed throughout the paper, along with other typographical errors. Please review and correct them.
Response: We decided to use the full terms and removed the abbreviations.
Comment 7: Discussion To enhance this section, I recommend adding a newly created summary table for the literature review, as mentioned in the discussion paragraph. This table will help streamline the discussion and reduce the need for lengthy descriptions. Additionally, the manuscript currently lacks a section addressing the limitations of the study, which should be included.
Response: We followed the reviewer’s advice and added a summary table of related studies on page 12; however, the table is very long and difficult to format and fit in one page. We did our best to provide the optimum formatting.
We also added a new section titled Limitations on page 16, Lines 449 to 461 and improved the conclusion.
Reviewer 2 Report
Comments and Suggestions for Authors
This study analyzed 606 cases of uveitis over 13 years, focusing on the etiology, anatomical localization, and demographic factors. The results showed that anterior uveitis (AU) was the most common type, with HLA-B27-associated uveitis (32.5%) and viral uveitis (16.8%) being the most frequent causes. The study also found that uveitis was more prevalent in patients aged 19-65, with viral uveitis more common in those over 65. The findings highlight the significant correlation between etiology and anatomical localization, with posterior uveitis (PU) showing the highest proportion of clarified etiologies. Identifying the cause of uveitis is crucial for effective treatment and clinical decision-making. There are some major comments.
- The study provides valuable insights into the epidemiology and etiology of uveitis, but it would benefit from a clearer explanation of the limitations of the data, especially regarding the potential biases of a single-center study.
- The authors didwell in categorizing uveitis types and correlating them with etiology, but further clarification on how idiopathic cases were differentiated from others would enhance the reliability of the findings.
- The article discussed the consistency and differences between the research findings and relevant studies from other regions, including Europe, the Middle East, Egypt, India, and South America. However, there is a lack of comparison with the East Asia. East Asia has a large population, and related studies have found an association between uveitis in China and systemic diseases (PMID: 32188681). The correlation between this study and the findings of those studies can be further discussed.
- The paper would benefit from a more detailed discussion on how the results can be applied to clinical practice, particularly in terms of diagnostic approaches for different etiologies of uveitis.
- The statistical analysis is well-conducted; however, it would be helpful to include more details on the diagnostic criteria used for rare etiologies to ensure transparency in the methodology.
- In discussing the etiology of uveitis, it is recommended to conduct a more in-depth discussion.For instance, current research suggested that the IL-23/IL-17 pathway plays a crucial role in both autoinflammatory and autoimmune uveitis. (PMID: 37734442) These types of content could be added to the background to enrich the your paper.
Author Response
REVIEWER 2
Overall
This study analyzed 606 cases of uveitis over 13 years, focusing on the etiology, anatomical localization, and demographic factors. The results showed that anterior uveitis (AU) was the most common type, with HLA-B27-associated uveitis (32.5%) and viral uveitis (16.8%) being the most frequent causes. The study also found that uveitis was more prevalent in patients aged 19-65, with viral uveitis more common in those over 65. The findings highlight the significant correlation between etiology and anatomical localization, with posterior uveitis (PU) showing the highest proportion of clarified etiologies. Identifying the cause of uveitis is crucial for effective treatment and clinical decision-making. There are some major comments.
Response: We thank the reviewer for providing his/her opinion on our manuscript with valuable suggestions. We have addressed them in this point-by-point response.
Comment 1: The study provides valuable insights into the epidemiology and etiology of uveitis, but it would benefit from a clearer explanation of the limitations of the data, especially regarding the potential biases of a single-center study.
Response: Thank you for this suggestion. We have added a Limitations section on page 16 of the manuscript, where the reviewer’s comment has been incorporated.
Comment 2: The authors did well in categorizing uveitis types and correlating them with etiology, but further clarification on how idiopathic cases were differentiated from others would enhance the reliability of the findings.
Response: We have added an explanation on page 4, Lines 168-171, and included it in the Limitations.
Comment 3: The article discussed the consistency and differences between the research findings and relevant studies from other regions, including Europe, the Middle East, Egypt, India, and South America. However, there is a lack of comparison with the East Asia. East Asia has a large population, and related studies have found an association between uveitis in China and systemic diseases (PMID: 32188681). The correlation between this study and the findings of those studies can be further discussed.
Response: Unfortunately, access to the full version of this article is paid, which did not allow us to include it in the reviewed literature.
Comment 4: The paper would benefit from a more detailed discussion on how the results can be applied to clinical practice, particularly in terms of diagnostic approaches for different etiologies of uveitis.
Response: We have elaborated on this issue on page 15, Lines 439-441; page 16, Lines 442 to 447.
Comment 5: The statistical analysis is well-conducted; however, it would be helpful to include more details on the diagnostic criteria used for rare etiologies to ensure transparency in the methodology.
Response: In the Methods section, we provided additional clarifications on the diagnostic criteria for rare diagnoses, page 4, Lines 150-168.
Comment 6: In discussing the etiology of uveitis, it is recommended to conduct a more in-depth discussion. For instance, current research suggested that the IL-23/IL-17 pathway plays a crucial role in both auto inflammatory and autoimmune uveitis. (PMID: 37734442) These types of content could be added to the background to enrich the your paper.
Response: Thank you for suggesting the above study, which presents a detailed analysis of the etiology, pathogenesis, clinical findings, and therapeutic approach in patients with Behçet's disease. However, in our geographic region and population, the disease is a rare cause of uveitis, which is why we have not examined this nosological entity in such depth.
Reviewer 3 Report
Comments and Suggestions for Authors
Dear Authors!
Thank you for the opportunity to review your manuscript
Uveitis is one of the most frequent cause of blindness and analysis of the uveitis reasons and the outcomes is very actual.
Authors provided the long-term analysis of more than 6 hundreds of the uveitis patients to assess the demography, epidemiology and cause.
The methods of the study are provided in details
The results are clear and the Authors provided the detailed analysis of the studied population
The discussion contains the comparison of the results and the previously published literature
I have several suggestions
1) please use only one digits in decimals for the continuous variables
2) Please provide the more detailed statistical statements about the check of the distribution and use the appropriate methods of the analysis
3) The detailed limitations sections is required. The main problem that the determination of the uveitis etiology is very difficult, especially for the infections. Immune-mediatied uveitis is often misdiagnosed as infectious.
Author Response
March 3rd, 2025
Dear REVIEWER 3,
Thank you very much for the insightful comments and suggestions you have provided on our manuscript titled Epidemiology of Uveitis from a Tertiary Referral Hospital in Bulgaria over 13-Year Period. We have tried to implement all of them in the revision process. We have highlighted in yellow all sections, where additional information was provided or suggested revisions were done. All language edits, including grammar, word choice, rephrasing, and punctuation are given in red color. Here follows a point-by-point description of the revisions.
Thank you for the opportunity to review your manuscript
Uveitis is one of the most frequent cause of blindness and analysis of the uveitis reasons and the outcomes is very actual.
Authors provided the long-term analysis of more than 6 hundreds of the uveitis patients to assess the demography, epidemiology and cause.
The methods of the study are provided in details
The results are clear and the Authors provided the detailed analysis of the studied population
The discussion contains the comparison of the results and the previously published literature
Response: We thank the reviewer for the reassuring feedback. Your opinion is greatly appreciated.
I have several suggestions
Comment 1: Please use only one digits in decimals for the continuous variables
Response: We have fixed this.
Comment 2: Please provide the more detailed statistical statements about the check of the distribution and use the appropriate methods of the analysis
Response: Our data has only one continuous variable, age. We added an explanation for how normality was checked on page 5, Lines 201 -203. We believe that the statistical methods were appropriate for the type of data, which was mostly categorical and described by frequencies and percentages.
Comment 3: The detailed limitations sections is required. The main problem that the determination of the uveitis etiology is very difficult, especially for the infections. Immune-mediatied uveitis is often misdiagnosed as infectious.
Response: Thank you for this suggestion. We have implemented your points in the added section Limitations on page 16, Lines 449 to 461 and improved the conclusion.
Round 2
Reviewer 3 Report
Comments and Suggestions for Authors
Dear Authors!
Thank you for the revised version of the manuscript
I have no additional comments
Good luck!